# Characterization of A Staphylococcal Food Poisoning Outbreak in A Workplace Canteen during the Post-Earthquake Reconstruction of Central Italy

**DOI:** 10.3390/toxins10120523

**Published:** 2018-12-06

**Authors:** Fabrizia Guidi, Anna Duranti, Silvia Gallina, Yacine Nia, Annalisa Petruzzelli, Angelo Romano, Valeria Travaglini, Alberto Olivastri, Vincenzo Calvaresi, Lucia Decastelli, Giuliana Blasi

**Affiliations:** 1Istituto Zooprofilattico Sperimentale Umbria e Marche “Togo Rosati”, via G. Salvemini 1, 06126 Perugia, Italy; a.duranti@izsum.it (A.D.); a.petruzzelli@izsum.it (A.P.); g.blasi@izsum.it (G.B.); 2National Reference Laboratory for Coagulase Positive Staphylococci - Istituto Zooprofilattico Sperimentale del Piemonte, Liguria e Valle d’Aosta, via Bologna 148, 10154 Torino, Italy; Silvia.Gallina@izsto.it (S.G.); Angelo.Romano@izsto.it (A.R.); Lucia.Decastelli@izsto.it (L.D.); 3European Union Reference Laboratory for Coagulase Positive Staphylococci - Laboratory for Food Safety, Anses, Université Paris-Est, F-94700 Maisons-Alfort, France; yacine.NIA@anses.fr; 4Presidio Ospedaliero C. e G. Mazzoni, Azienda Sanitaria Unica Regione Marche, Area Vasta n°5, via Degli Iris 1, 63100 Ascoli Piceno, Italy; valeria.travaglini@sanita.marche.it; 5Dipartimento di Prevenzione, Azienda Sanitaria Unica Regione Marche, Area Vasta n°5, Viale Marcello Federici, 63100 Ascoli Piceno, Italy; alberto.olivastri@sanita.marche.it (A.O.); vincenzo.calvaresi@sanita.marche.it (V.C.)

**Keywords:** *Staphylococcus aureus*, Staphylococcal Enterotoxins, Food poisoning, Outbreak investigation, PFGE, biotyping

## Abstract

In summer 2017, a foodborne outbreak occurred in Central Italy, involving 26 workers employed in the post-earthquake reconstruction. After eating a meal provided by a catering service, they manifested gastrointestinal symptoms; 23 of them were hospitalized. The retrospective cohort study indicated the pasta salad as the most likely vehicle of poisoning. Foods, environmental samples, and food handlers’ nasal swabs were collected. *Bacillus cereus* (*Bc*) and coagulase-positive staphylococci (CPS) including *S. aureus*, together with their toxins, were the targets of the analysis. CPS, detected in all the leftovers, exceeded 10^5^ CFU/g in the pasta salad, in which we found Staphylococcal Enterotoxins (SEs) (0.033 ng SEA/g; 0.052 ng SED/g). None of the environmental and human swabs showed contamination. We characterized 23 *S. aureus* from foods. They all belonged to the human biotype, showed the same toxigenic profile (*sea*, *sed*, *sej*, and *ser* genes), and had the same Pulsed Field Gel Electrophoresis (PFGE) pattern; none of them harbored *mecA* or *mupA* genes. We also detected *Bc* contamination in the pasta salad but none of the isolates harbored the *ces* gene for the emetic toxin cereulide. The EU Reference Laboratory for CPS confirmed the case as a strong-evidence outbreak caused by the ingestion of SEs produced by a single strain of *S. aureus* carried by the same human source. This outbreak was successfully investigated despite the emergency situation in which it occurred.

## 1. Introduction

*Bacillus, Clostridium*, and *Staphylococcus* bacterial toxins are important causes of foodborne disease outbreak (FBO); in 2016, these agents represented the second most frequent cause of FBOs in Europe, typically occurring in canteens, workplace catering, schools, and hospitals [1]. Despite increased vigilance by health authorities, these still occur in Italy, as demonstrated by a recent staphylococcal outbreak from a cream dessert [2].

*Staphylococcus aureus*, commonly harbored by about 30–60% of healthy people in the skin and respiratory mucosa, is a frequent source of FBOs [3]. Many of these coagulase-positive strains are enterotoxigenic with the risk of contaminating ready-to-eat (RTE) foods with heat-resistant Staphylococcal Enterotoxins (SEs). Thus, food handlers must comply with the fundamental principles of good hygienic practices through proper training to avoid contamination and subsequent growth of the organism on RTE food [4,5].

A wide variety of SEs causing staphylococcal food poisoning outbreaks (FPO) in humans have been reported [6]. The symptoms of SEs—which mainly consist of nausea, emesis, abdominal cramps, and diarrhea—occur 2–8 h after the ingestion of contaminated food [7]. 

More than 20 SEs and SE-like toxins (SEls) have been described in the literature over more than 50 years, whereas only five enterotoxins of type A (SEA), B (SEB), C (SEC), D (SED), and E (SEE) can be detected using commercial immune-assays or methods developed in-house. Due to the lack of suitable antibodies, no validated method is available for the detection and quantification of enterotoxin types other than SEA-SEE, such as SEs of type G (SEG), H (SEH), and I (SEI), which are known to be a risk for consumers [8]. For this purpose, many immunological assays are under development [9].

Several foodborne outbreaks in Europe have shown a typical SE symptomatology but no enterotoxins of types SEA to SEE could be detected. However, genes such as *seg, seh, sei, sem, sen,* and *ser* were detected using polymerase chain reaction (PCR), probably indicating the presence of toxins of types SEG, SEH, and SEI [3].

Moreover, during an FPO, *S. aureus* isolates can be differentiated into host-specific (human, bovine, and ovine) or non-host-specific biotypes; this information plays a fundamental role in the investigation of the source of food contamination [10].

The aim of this study was to describe the epidemiological and laboratory investigations of a recent FBO that occurred in a corporate canteen (summer 2017) involving a company of workers employed in the post-earthquake reconstruction in Central Italy. Measures to prevent the recurrence of similar episodes are proposed, in particular the use of refrigerated transport with a temperature monitoring system.

### Background

One day in summer 2017 (day 1), at 10:00 p.m., the Emergency Department in Ascoli Piceno (AP), Marche Region (Italy), notified the local health authority (LHA) of a suspected FBO. Shortly after dinnertime, many workers employed in the post-earthquake reconstruction in Arquata del Tronto (AP) manifested abdominal pain, nausea, and emesis, in a few cases with fever and/or diarrhea; 23 of them were admitted to the emergency room. All the patients had eaten the meal provided by a catering service that produces meals for a wide range of canteens.

## 2. Results

### 2.1. Epidemiological Investigation

In the preliminary phase of the epidemiological investigation, we collected data concerning the meal and the catering service. The dinner courses were a selection of mixed sliced meat containing ham, pork loin, and salami, a pasta salad, and boiled carrots served as a side dish. A total of 56 single-portion meals were delivered but only 48 workers ate one of them.

The catering service prepares single-serve food packaging at the centralized kitchens; the meal delivery is subcontracted to another company whose vehicles are equipped with insulated food bins. Two cooks, apparently in good health, prepared and packaged the meals.

Only 36 workers, all male, were traced and surveyed, belonging to the study cohort; no active case finding was carried out. A positive case was defined as any person, hospitalized or not, with an acute onset of vomiting in a six-hour period after consuming the canteen meal for dinner on day 1. The workers ate the meal between 6:30 p.m. and 8:20 p.m. A total of 26 of them developed the symptoms according to the case definition, and 23 were hospitalized. The median age of the cases was 33 years (range 19–56). The onset of symptoms occurred between 7:30 p.m. on day 1 and 2:00 a.m. on the following day, with a median duration of 8 h. The median incubation time was about 2 h. Among the affected workers (*n* = 26), symptoms always included emesis with abdominal pains (24; 92%), diarrhea (21; 81%), fever (11; 42%), and headache (9; 35%).

### 2.2. Statistical Analysis

We calculated the relative risk (RR) values for all the dinner courses except for the pasta salad, for which we used attributable risk (RA).

The results from the retrospective cohort study (Table 1) implicated the pasta salad as the most likely vehicle of food poisoning. Indeed, the RA of this food was quite high (72%) and, in addition, the IC 95% for all other foods’ RR included the null value of 1, indicating that the difference in risk between those who ate them and those who did not was not statistically significant.

### 2.3. Laboratory Investigation

#### 2.3.1. Sample Collection

In the early hours of the day following the clinical manifestations, the Sanitary Inspectors of the LHA collected three samples of leftovers from the dinner (mixed sliced meat containing ham, pork loin, and salami, pasta salad, and boiled carrots) sampled from the unconsumed single-portion meals.

Forty-eight hours later, inspectors went to the company’s kitchen and sampled four of the ingredients used to prepare the dinner (baked ham, Emmental cheese, Provolone cheese, and mixed milk cheeses).

Two environmental samples were collected from the slicer and the steel food container sanitized by the employees.

Two nasal swabs, one from each food handler, were also collected.

#### 2.3.2. Microbiological and Molecular Analysis

Based on the symptoms, the food investigations were oriented towards *Bacillus cereus* (*Bc*) and coagulase-positive staphylococci (CPS) including *S. aureus*. The environmental and human swabs, instead, were tested for CPS and *S. aureus* contamination.

The microbiological results regarding the foodstuffs are shown for both CPS and *Bc*, in Table 2.

CPS contamination was detected in all the leftover samples and exceeded 10^5^ CFU/g in the pasta salad.

*Bc* contamination was detected only in the pasta salad sample but, as the colony count for all the inoculated dilutions produced a number greater than 150, the result was expressed as “more than 150,000 CFU/g” and it was not possible to define the exact value [11].

None of the environmental and human swabs showed CPS and *S. aureus* contamination.

Qualitative immunoassay detection indicated the presence of SEs (types SEA to SEE) in the pasta salad. The quantitative enzyme-linked immunosorbent assay (ELISA) confirmed the sample contamination and the presence of SEA and SED at a concentration of 0.033 ng/g and 0.052 ng/g, respectively (Table 2).

A total of 23 CPS isolates, all identified as *S. aureus*, were characterized: 13 from the pasta salad, 5 from the mixed slices, and 5 from the boiled carrots. They showed the same toxigenic molecular profile, harboring *sea*, *sed*, *sej*, and *ser* genes (Figure 1); none of them tested positive for *mecA* or *mupA* genes.

All the *S. aureus* isolates had the same PFGE restriction pattern (98.52% similarity; Figure 2) and belonged to the human biotype.

Furthermore, five presumptive *Bc* strains isolated from the pasta salad were confirmed as belonging to the Bacillus cereus group by multiplex PCR, but none of them harbored *ces* gene for the emetic toxin cereulide.

## 3. Discussion

In the case described in this paper, the rapid onset of the illness after the consumption of a meal and the type of symptoms suggested it was food poisoning caused by bacterial toxins; the results from the retrospective cohort study indicated the pasta salad as the most likely vehicle.

The toxigenic agents potentially involved were CPS, including *S. aureus*, and *Bc. Clostridium perfringens* was ruled out because, despite some patients having diarrhea, the main symptom that occurred was emesis, which is not associated with this pathogen; moreover, none of the courses were a typical source of this agent.

One or more of the following criteria generally confirm the diagnosis of Staphylococcal food poisoning: (1) the detection of SEs in food leftovers; (2) the isolation of the same *S. aureus* strain from both patient vomitus and stool samples and food leftovers; and/or (3) the recovery of more than 10^5^ CFU/g of CPS and the identification of *S. aureus* from food leftovers [8].

During the investigation, we could not check the second condition, as we could not obtain vomit samples from cases and stool samples were not analyzed for CPS because this analysis is not routinely performed by the hospital laboratories. In any case, criteria (1) and (3) were met in the same food sample, identifying the outbreak as a case of poisoning caused by SEs; the isolation of nonproductive toxin *Bc* strains would exclude the involvement of *Bc* and supported this hypothesis.

The use of polystyrene boxes for the transport of meals without refrigeration conditions was identified as the main hazard of the supply chain that allowed CPS contamination to reach such a high level. Therefore, the use of refrigerated transport with a temperature monitoring system was suggested to prevent such outbreaks. However, the advanced and unopened single-serve packages were left at room temperature after the dinner, which may have contributed to the amount of CPS being reached. It would have been useful to detect the food temperature at the sampling time.

As the European Union Reference Laboratory (EURL) for CPS stated that the enterotoxin recovery of the confirmatory method is approximately 50%, the effective amount in the sample could be doubled if compared to the detected one. Therefore, the amount of SEs in a pasta salad portion (about 250 g) was estimated to range from about 8.3 ng to 16.5 ng for SEA, and from 13 ng to 26 ng for SED. The effective dose of SEs causing intoxication in humans is still unclear; Guillier et al. estimated that, for SEA, the 95% lower confidence limit of the Benchmark Dose, inducing effects in 10% of the exposed population, was 6.1 ng [12]. This value is very close to the one found in this outbreak. However, many previous studies detected a higher amount of SEs inducing symptoms; if we consider the contribution of both toxins, the resulting value is in accordance with Denayer et al. [2,13,14,15].

On the other hand, about 5% of Staphylococcal food poisoning might have been due to recently identified SEs, which are not routinely analyzed [16]. Therefore, we cannot exclude that other SEs contributed to the outbreak. Indeed, the *S. aureus* isolates—in addition to harboring *sea* and *sed* genes, consistently with the detection of the relative SEA and SED toxins—also showed the targets *ser* and *sej*. These genes, encoding for the SEs of types R (SER) and J (SEJ) respectively, are the most frequently encountered in Italy and have been detected in many FPOs also abroad [17]. In addition, for SER the emetic activity has been demonstrated [18].

Also noteworthy is the fact that out of the 24 SEs genes identified to date, only 11 can be detected using validated multiplex PCR assays, while the others remain unnoticed.

All the isolates had the same toxigenic profile, shared identical PFGE patterns, and belonged to human serovar. These findings demonstrated the existence of a human carrier as the most probable source of the outbreak. Despite the fact that no strains were isolated from the human swabs, the biotyping method supported the hypothesis of a food-handler reservoir. The lack of strain isolation from the nasal swabs could probably be due to the fact that actual *S. aureus* screening programs require a swab specimen from the anterior nares only, with additional sampling from the throat being considered to be unnecessary to increase sensitivity. Indeed, despite throat carriers of *S. aureus* being likely to carry it in the nares as well, colonization of the throat but not of the nares may be more common than is currently acknowledged [19]. Furthermore, the two company food handlers could carry *S. aureus* even at the skin level, and although they have the skills and knowledge to handle food safely, human handling errors are possible and frequent.

Considering this, it would have been useful to collect hand swabs from the handlers.

Despite the above limits, the present investigation provided sufficient findings to suggest that the FBO was due to the ingestion of SEs. The use of both phenotypic and genotypic characterization techniques on *S. aureus* isolates from specimens was a fundamental tool for the investigation. In particular, molecular biology showed that the contamination had the same clonal origin, and biotyping proved its human source. We reported microbiological, immunological, and characterization results to the EURL for CPS, which defined this outbreak as a strong-evidence outbreak with SEs as the causative agent. The case was reported to the European Food Safety Authority (EFSA) for its insertion in “The European Union summary report on trends and sources of zoonoses, zoonotic agents and food-borne outbreaks in 2017”.

## 4. Conclusions

In the post-earthquake context, in which the scale of health priorities changes, epidemiological investigations become very difficult because of the many logistical problems in terms of road conditions, the availability of suitable vehicles for inspections, tracing cases, and information flow. A consolidated collaboration between local health authorities is fundamental for sharing resources and skills.

The quick identification of this Staphylococcal food poisoning is the result of a close and effective cooperation between the local health authority, the laboratory network of Istituto Zooprofilattico Sperimentale of Umbria and Marche “Togo Rosati”(IZSUM), the Italian National Reference Laboratory and the EURL for CPS, which provided scientific and technical support.

Food poisoning outbreaks such as the one described here have a great impact on society and highlight the fragility of food safety measures. Nevertheless, they often go unnoticed because of the difficulty of tracing the source of infection of patients affected and, above all, of tracing the foods involved and the source of their contamination [20]. This paper describes one of the few cases in which a foodborne outbreak was successfully investigated and classified as a strong-evidence staphylococcal food poisoning with the identification of both the food involved and the source of contamination [2,4]. Moreover, by notifying the case to the EFSA, we contributed to the European Union Food-borne Reporting System, directed at reporting FBO and assessing their trends.

This study demonstrates how the timely sampling of foods and the quick distribution of questionnaires to patients are essential means for successfully investigating an outbreak. However, the need for guidelines to follow emerged in order to better perform sample collections from the environment, patients, and food handlers. 

The combination of the epidemiological investigation and microbiological, immunological, and molecular analyses plays a fundamental role in the elucidation of FBOs, and once the microbial agent has been identified and isolated, the use of different characterization methods represents an effective approach to understand the origin and dynamics of the contamination.

Compliance with good hygiene practices by food handlers and maintaining cold meals at refrigerated temperatures proved to be essential. In particular, in emergency conditions such as a post-earthquake reconstruction, it is even more difficult to ensure food safety and hygiene, therefore the development of shared and standardized operative protocols is necessary.

## 5. Materials and Methods

### 5.1. Study Design

We postulated that the meal served at the corporate canteen was the source of the outbreak and that the rapid onset and symptoms of the illness were suggestive of an FBO due to bacterial toxins. We designed a retrospective cohort study to test this hypothesis. The study population consisted of workers (*n* = 48) who had dined on all or some of the courses of the canteen meal on the day on which the symptoms occurred (day 1). A positive case was defined as any person, hospitalized or not, with an acute onset of vomiting in a six-hour period after consuming the canteen meal for the dinner on day 1.

### 5.2. Epidemiological Investigation

Sanitary inspectors from the local health authority prepared a questionnaire for the workers recording demographic information, characteristics of the illness, and information about the foods consumed.

A cohort of 36 completed questionnaires formed the database for the statistical analysis.

The food-specific attack rate (AR%) for each dinner course was calculated both for those who ate the specific food and for those who did not. The two attack rates were then compared with each other, dividing them to find the relative risk (RR) or subtracting them to find the attributable risk (RA). The RA was used if the dividend value in the RR was zero [21]. Epi-Info Software Version 7.2 (Centers for Disease Control and Prevention, Atlanta, GA, USA, 2017) was used to calculate these rates and their 95% Confidence Intervals.

### 5.3. Laboratory Investigation

#### 5.3.1. Sample Collection

The sanitary inspectors from the local health authority collected samples of leftovers from the dinner (mixed sliced meat containing ham, pork loin, and salami, pasta salad, and boiled carrots) from the unconsumed single-portion meals. The samples were transported under refrigeration (1–8 °C) to Fermo (about 100 Km from Arquata del Tronto) at the Istituto Zooprofilattico Sperimentale of Umbria and Marche “Togo Rosati” (IZSUM) where they were analyzed. Forty-eight h later, inspectors went to the company’s kitchen and sampled four of the ingredients used to prepare the dinner (baked ham, Emmental cheese, Provolone cheese, and mixed milk cheeses).

Two environmental samples were collected from the slicer and the steel food container sanitized by the employees.

According to EC Regulation 882/2004, the competent authorities used appropriate sampling procedures to guarantee the right of food business operators to apply for a supplementary expert opinion [22].

Nasal swabs collected from the food handlers were transported to the C. e G. Mazzoni Hospital for laboratory analysis (Azienda Sanitaria Unica Regione Marche (ASUR)-Area Vasta n°5-AP) in Ascoli Piceno (about 30 km from Arquata Del Tronto) for the detection of *S. aureus*.

#### 5.3.2. Microbiological and Molecular Analysis

Food samples were analyzed for the enumeration of CPS and *Bc* using standards UNI EN ISO 6888-2:2004 and UNI EN ISO 7932:2005, respectively [23,24].

The environmental samples were investigated for CPS enumeration only, according to ISO 18593:2004 (except Chapters 6 and 7) and UNI EN ISO 6888-2:2004 protocol [23,25].

Food specimens were also tested for staphylococcal enterotoxins, according to the official European Screening Method (ESM) which can simultaneously detect SEA to SEE in food matrices without differentiating the five SEs (ESM of the European Union Reference Laboratory for “Coagulase positive staphylococci, including *Staphylococcus aureus*” (EURL for CPS), VER 5:2010) [26,27]. The EURL for CPS confirmed the positive samples, using an in-house Enzyme-linked Immunosorbent Assay (ELISA) method [28]. Briefly, double-sandwich ELISA types were used for SEA, SEC, and SED, whereas a single-sandwich type was used for SEB. Specific commercially available antibodies (Toxin Technology, Sarasota, FL, USA) were used as coating (references SLAI101, SLBI 202, SLCI 111, SLDI 303) and probing antibodies (references LAI101, LBC 202, LCI 111, and LDI 303). Immunoglobulins coupled to horseradish peroxidase (goat anti-rabbit antibodies coupled to peroxidase) were used for detection purposes using a colorimetric measurement at 405/630 nm after the addition of the substrate, ABTS-H_2_O_2_ (KPL). SEs standards were purchased from Toxin Technology (Sarasota, FL, USA (batch 120794A for SEA, 61499B1 for SEB, 113094C2 for SEC, and 12802D for SED)).

Nasal swabs were directly inoculated onto Mannitol salt agar plates (Becton Dickinson). Diagnosis of *S. aureus* was performed using the clinical laboratory’s routine method with incubation at 35 °C from 24 to 48 h and confirmation of suspected colonies using Gram stain, 3% catalase testing, and the Staph Latex agglutination assay (DID). CPS isolates were confirmed as *S. aureus* using a multiplex PCR assay based on the simultaneous detection of 16s rRNA (genus specific) and *nuc* (specific for *S. aureus*) genes [29]. The assay targets also *mecA* and *mupA* genes, encoding for methicillin and mupirocin resistance, respectively.

*S. aureus* strains were tested for SEs genes, using two multiplex PCR assays, according to the EURL for CPS method; one assay targets *sea*, *seb*, *sec*, *sed*, *see*, and *ser* genes, while the other targets *seg*, *seh*, *sei*, *sej*, and *sep* [30,31].

Further molecular characterization of the *S. aureus* strains was performed using Pulsed Field Gel Electrophoresis (PFGE) according to the EURL for the CPS method [32]. The restriction enzyme used was a cut smart *SmaI* (New England BioLabs^®^, Ipswich, MA, USA). Restriction fragments were separated using the CHEF-Mapper^®^ electrophoresis system (Bio-Rad Laboratories, Hercules, CA, USA). Gel images were sent to the Italian National Reference Laboratory for CPS (IT-NLR, Turin) and analyzed using BioNumerics software vers. 7.6 (Applied-Maths, Sint-Martens-Latem, Belgium). An unweighted pair group method with an arithmetic mean (UPGMA) clustering algorithm was used to create the related dendrogram, and the genetic similarity between isolates was calculated using the Jaccard coefficient with the band optimization and tolerance set at 0.5% and 2%, respectively.

*S. aureus* strains were also biotyped to identify their source (human, animal, or non-host-specific), using the simplified scheme which employs four discriminative tests: the production of staphylokinase and β-haemolysin, coagulation of bovine plasma within 6 h, and the type of growth on crystal violet agar [33,34].

Presumptive *B. cereus* strains were confirmed using a multiplex PCR assay targeting *gyrB* and *cer* genes, the first specific of the *Bacillus cereus* group and the second coding for the emetic toxin cereulide [35,36].

## Figures and Tables

**Figure 1 toxins-10-00523-f001:**
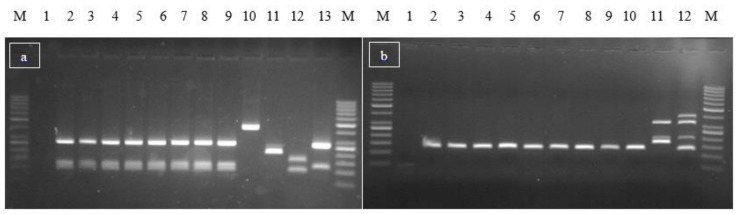
Multiplex PCR for SEs genes: results. (**a**) Detection of *sea*, *seb*, *sec*, *see*, and *ser* genes: lane 1: negative control; lanes 2–9: samples; lanes 10, 11, 12, and 13: positive controls for *sec* (451 bp), *see* (213 bp), *sea* (102 bp), *seb* (164 bp), *sed* (278 bp), and *ser* (123 bp). (**b**) Detection of *seg*, *seh*, *sei*, *sej*, and *sep* genes: lane 1: negative control; lanes 2–10: samples; lane 11: positive control for *seg* (198 bp), *seh* (173 bp), and *sei* (328 bp); lane 12: positive control for *seg, seh*, *sej* (131 bp), and *sep* (396 bp). M, 100 bp size Marker.

**Figure 2 toxins-10-00523-f002:**
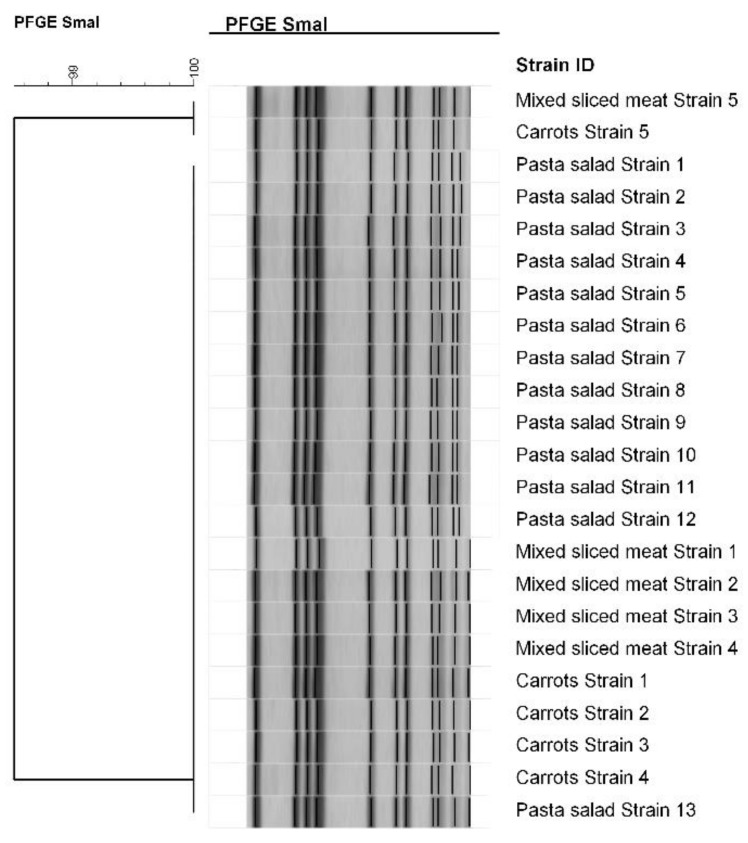
Dendrogram generated using the Dice/Unweighted pair group method using arithmetic averages (UPGMA) analysis. Determination of the similarity between the PFGE profiles (*SmaI*) of the *S. aureus* isolates.

**Table 1 toxins-10-00523-t001:** Food-specific Attack Rates (AR) (%) for exposed (cases exposed/total exposed) and unexposed (cases not exposed/total not exposed), Risk Ratios (AR exposed/AR not exposed—RR), and Attributable Risk (%) (AR exposed/AR not exposed—ra).

Foods	Exposed (Ate the Specific Food)			Not Exposed (Did Not Eat the Specific Food)			RR	RA	IC 95%	
	Cases	Total	AR	Cases	Total	AR			Lower	Upper
1. Pasta salad	26	36	72%	0	0	n.a.	n.a.	72%	57%	87%
2. Carrots	11	16	68.75%	15	20	75.00%	0.92	/	0.60	1.31
3. Ham	21	29	72.41%	5	7	71.43%	1.01	/	0.60	1.71
4. Loin	18	25	72.00%	8	11	72.73%	0.99	/	0.64	1.53
5. Salami	22	31	70.97%	4	5	80.00%	0.89	/	0.54	1.45

IC 95%—Confidence Interval (95%); n.a.—not applicable.

**Table 2 toxins-10-00523-t002:** Laboratory analysis results of food samples. CFU: colony forming unit; CPS: coagulase-positive staphylococci.

Food sample	Sample Unit Number	CPS Enumeration (CFU/g)	*Staphylococcal* Enterotoxin (Detection)	*Staphylococcal* Enterotoxins (Quantification)	*Bacillus cereus* Enumeration (CFU/g)
Mixed sliced meat	Single unit	1500	Not done	/	<100
Pasta salad	Single unit	99,000,000	Detected	SEA 0.033 ng/g; SEB < LoQ; SEC < LoD; SED 0.052 ng/g	>150,000
Boiled carrots	Single unit	610	Not found	/	<100
Baked ham	1	<10	Not found	/	100
	2	<10	Not found	/	<100
	3	<10	Not found	/	Detected <400
	4	<10	Not found	/	<100
	5	<10	Not found	/	<100
Emmental cheese	1	<10	Not found	/	<100
	2	<10	Not found	/	<100
	3	<10	Not found	/	<100
	4	<10	Not found	/	<100
	5	<10	Not found	/	<100
Provolone cheese	1	<10	Not found	/	<100
	2	<10	Not found	/	Detected <400
	3	<10	Not found	/	<100
	4	<10	Not found	/	<100
	5	<10	Not found	/	<100
Mixed cheese	1	<10	Not found	/	<100
	2	<10	Not found	/	<100
	3	<10	Not found	/	<100
	4	<10	Not found	/	<100
	5	<10	Not found	/	<100

LoQ: limit of quantification; LoD: limit of detection. LoDs were estimated as follows: 0.016 ng/mL for SEA, 0.125 ng/mL for SEB, 0.009 ng/mL for SEC, and 0.042 ng/mL for SED. LoQs were estimated as follows: 0.066 ng/mL for SEA, 0.0641 ng/mL for SEB, 0.033 ng/mL for SEC, and 0.135 ng/mL for SED. Conversion factor for the quantification: 1 ng/g corresponds to ≅ 4.9 ng/mL.

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
