# Peer review of "Characterization of A Staphylococcal Food Poisoning Outbreak in A Workplace Canteen during the Post-Earthquake Reconstruction of Central Italy"

_toxins, 2018, doi:10.3390/toxins10120523_

Round 1
Reviewer 1 Report
Thank you for the opportunity to review the review manuscript entitled, “Characterization of a staphylococcal food poisoning outbreak occurred in a workplace canteen during the post-earthquake reconstruction of central Italy.” The authors examined an outbreak of S. aureus gastroenteritis related to consumption of pasta salad in July 2017, with an investigation of 36 subjects. They identified closely related (by PFGE) strains of S. aureus in food samples; the isolates carried sea, sed, sej and ser staphylococcal enterotoxin genes. The authors did test a food handler as well as environmental surfaces but identified no S. aureus in these tests. Bacillus cereus was also isolated in the pasta salad samples, but this species was found in low numbers and lacked the presence of the emetic toxin cereulide.
This outbreak report is remarkable for the clarity of the methods, the modifiable risk factor for growth of the causative bacterium, and the case-control study that demonstrated the very likely food that served as the vector. While there were no human vomitus samples or cultures of the hands of food handlers available to test for the presence of S. aureus or its toxins, which would confirm the likely agent, the investigation was certainly successful. While there are many S. aureus gastroenteritis outbreaks each year in Europe, this one is particularly interesting in that it represents the work of public health operating successfully in the difficult conditions after a natural disaster. The dedication of public workers in difficult conditions to maintain the health of the population is admirable and is a demonstration of the value of public health to prevent the spread of future infections by identifying simple, affordable, and effective interventions. In the case of this study, the authors found that maintenance of meals at appropriate temperatures during transport would likely prevent similar outbreaks in the future.
I have the following specific comments and questions for the authors, some of them minor:
1. Line 56. In general, I defer to the judgement of the editors, but it is advised that specific dates not be utilized in research publications in order to protect the identity of study subjects. Can the authors eliminate the specific dates that they use and just give the month and year (or better the season and year, “At 10 PM on a day (day 1) in the summer of 2017”)?
2. line 67. “took and eat their own” is not clear. Can the authors please rephrase this?
3. Table 1. Please replace “,” with “.” in the numbers of all columns.
4. Lines 132-222. The Discussion and Conclusions can be shortened by at least 25%. They include much repetition of the Results, and they can be made more concise (especially lines 149-169 and lines 197-222).
5. Line 204. “…because of to…” cannot be interpreted. Please revise this.
6. Lines 246-247. Where (what city/town) is the Istituto Zooprofilattico Sperimentale of Umbria and Marche? How far is it from the site of the outbreak?
7. Lines 255-256. Where (what town) is the C. e G. Mazzoni Hospital? How far is it from the site of the outbreak?
8. Lines 296-299. This “biotyping” system is relevant, but it would be much more interesting and provide more generalizable knowledge if the authors could perform spa typing or multilocus sequence typing on at least one of the isolates from this outbreak. There have been many studies performed in Italy of livestock S. aureus (MSSA) carriage with genotyping in Italy (e.g., Vitale M, et al. Foodborne Pathog Dis, 2018; 15:177-185; Mizzilli M, et al. Res Vet Sci, 2015; 103:54-9; Vitale M, et al. Foodborne Pathog Dis 2018; 15:177-185; Carfora V, et al. J Dairy Sci 2016; 99:4251-8; Azara E, et al. Vet Microbiol 2017; 205:53-56). There have also been many studies of human MSSA carriage in Italy with genotyping (e.g., Franco A, et al. J Antimicrob Chemother 2011; 66:1231-5; Tinelli M, et al. Emerg Infect Dis 2009; 15:250-7; Grundmann H, et al. PLOS Medicine 2010; 7(1): e10000215; Vitale M, et al. Foodborne Pathog Dis, 2018). It would be quite relevant and increase the value of the manuscript to determine if the strain that caused the outbreak was 1) related to livestock strains in Italian food and/or livestock and 2) if the strain was related to other S. aureus foodborne outbreaks in recent years in southern Europe.
9. Can the authors comment on the impact of the fact that the outbreak investigation was carried out in the setting of the aftermath of a natural disaster? How did this impede the outbreak investigation or the response? Are there any lessons to be derived from this experience that may be useful to others in similar post-disaster environments?
Thank you again for the opportunity to review this interesting manuscript.
Author Response
Dear Reviewer,
Authors would like to thank you for the positive considerations and for showing esteem for their work and their dedication to maintaining public health.
Authors would like to thank you also for all useful comments and constructive suggestions, which help to improve the quality of this manuscript.
I have listed below, point-by-point, all changes and additions responding to your comments. The revisions on the manuscript have been clearly highlighted using different colors for each Reviewer and in particular red for Reviewers 1, blue for Reviewer 2 and green for Reviewer 3.
As requested by the Editor, the Key Contribution has been included in the revised version.
Point 1: Line 56. In general, I defer to the judgement of the editors, but it is advised that specific dates not be utilized in research publications in order to protect the identity of study subjects. Can the authors eliminate the specific dates that they use and just give the month and year (or better the season and year, “At 10 PM on a day (day 1) in the summer of 2017”)?
Au: Agreed and done.
Response 1: Authors would like to thank the Reviewer 1 for this useful comment, which helps to protect the identity of study subjects. The change suggested has been done in all the sentences in which the specific dates or the month appeared. The correction line by line are shown below:
· Line 6. “On July 2017” “In the summer 2017” is better.
· Line 51. “(July 20174)” replaced with “(Summer 2017)”
· Line 56. “July 15th, 2017, at 10 pm…” replaced with “One day in the summer 2017 at 10 pm (day 1)”
· Line 73. “15 July 2017” replaced with “day1”
· Line 76-77: “..occurred between 7.30 pm on day 1 and 2.00 am on the following day..”
· Line 98. “(July 2017)” has been deleted
· Line 103. “two nasal swabs, one from each food handlers were also collected” is better.
· Line 229.“ ..15 July 2017” replaced with “ on the day the symptoms occurred (day 1)”
· Line 231. “ ..15 July 2017” replaced with “day 1”
· Line 248. “(July 2017)” has been deleted
Point 2: line 67. “took and eat their own” is not clear. Can the authors please rephrase this?
Au: Agreed and done.
Response 2: “only 48 workers ate one of them..” is clearer.
Point 3: Table 1. Please replace “,” with “.” in the numbers of all columns.
Au: Agreed and done.
Point 4: Lines 132-222. The Discussion and Conclusions can be shortened by at least 25%. They include much repetition of the Results, and they can be made more concise (especially lines 149-169 and lines 197-222).
Au: Agreed and done.
Response 4: Authors agree that the period indicated from the Reviewer 1 is quite redundant and needs to be lightened. For this purpose the following revisions have been made:
· Lines 149-154. The period has been changed as follows: “The use of polystyrene boxes for the transport of meals without refrigeration conditions was identified as the main hazard of the supply chain that allowed CPS contamination to reach such a high level. Therefore, the use of refrigerated transport with temperature monitoring system was suggested to prevent such outbreaks. ”
· Lines 159-163. The period has been changed as follows ”Therefore the amount of SEs in a pasta salad ingested portion (about 250g) was estimated to range from 8.3 to 16.5 ng for SEA and from 13 to 26 ng for SED.”
· Lines 201-202. “ Setting up a multidisciplinary…” has been deleted
Point 5: Line 204. “…because of to…” cannot be interpreted. Please revise this.
Au: Agreed and done.
Response 5: Authors thank the Reviewer 1 for showing them the misprint, the correct sentence is “…because of the difficulty in tracing..”
Point 6: Lines 246-247. Where (what city/town) is the Istituto Zooprofilattico Sperimentale of Umbria and Marche? How far is it from the site of the outbreak?
Au: Agreed and done.
Response 6: Authors would like to thank the Reviewer 1 for suggesting to specify the location of the laboratory of the Istituto Zooprofilattico Sperimentale of Umbria and Marche that performed analysis. The information has been inserted as shown below:
Lines 246-247. “…the samples were transported under refrigeration (1-8°C) to Fermo (about 100 Km from Arquata Del Tronto) at the Istituto Zooprofilattico of Umbria and Marche where they were analyzed.
Point 7: Lines 255-256. Where (what town) is the C. e G. Mazzoni Hospital? How far is it from the site of the outbreak?
Au: Agreed and done.
Response 7: As in the previous point the authors agree with the Reviewer’s observation. The information has been inserted as shown below:
Lines 255-256. “….were transported to the C. e G. Mazzoni Hospital laboratory analysis (Azienda Sanitaria Unica Regione Marche (ASUR)- Area Vasta n°5-AP) in the city of Ascoli Piceno (about 30 Km from Arquata Del Tronto)…”
Point 8: Lines 296-299. This “biotyping” system is relevant, but it would be much more interesting and provide more generalizable knowledge if the authors could perform spa typing or multilocus sequence typing on at least one of the isolates from this outbreak. There have been many studies performed in Italy of livestock S. aureus (MSSA) carriage with genotyping in Italy (e.g., Vitale M, et al. Foodborne Pathog Dis, 2018; 15:177-185; Mizzilli M, et al. Res Vet Sci, 2015; 103:54-9; Vitale M, et al. Foodborne Pathog Dis 2018; 15:177-185; Carfora V, et al. J Dairy Sci 2016; 99:4251-8; Azara E, et al. Vet Microbiol 2017; 205:53-56). There have also been many studies of human MSSA carriage in Italy with genotyping (e.g., Franco A, et al. J Antimicrob Chemother 2011; 66:1231-5; Tinelli M, et al. Emerg Infect Dis 2009; 15:250-7; Grundmann H, et al. PLOS Medicine 2010; 7(1): e10000215; Vitale M, et al. Foodborne Pathog Dis, 2018). It would be quite relevant and increase the value of the manuscript to determine if the strain that caused the outbreak was 1) related to livestock strains in Italian food and/or livestock and 2) if the strain was related to other S. aureus foodborne outbreaks in recent years in southern Europe.
Response 8: The biotyping system was used only for identifying the origin of contamination and in particular its animal or human nature. It was only a method for solving and understanding the outbreak in a short time. However, Authors agree with the importance of a further characterization study on the isolates from the outbreak, and would to thank the Reviewer 1 for the interesting suggestion. Determining the correlation between the strain of the outbreak and other strains in Italian food or understanding if it was related with other S. aureus foodborne outbreaks in recent years in southern Europe, would be very useful for making epidemiological evaluations. For this purpose, Authors have included the isolates of this case in a further research project planned by the Italian National Reference Laboratory for CPS, which will consist of Whole Genome Sequencing of the strains isolated from different outbreaks. WGS methods allow access to a large amount of information on the sequenced strain.
Point 9: Can the authors comment on the impact of the fact that the outbreak investigation was carried out in the setting of the aftermath of a natural disaster? How did this impede the outbreak investigation or the response? Are there any lessons to be derived from this experience that may be useful to others in similar post-disaster environments?
Au: Agreed and done.
Response 9: In the post-earthquake scenario, in which the scale of health priorities changes, making an epidemiological investigation was very difficult because of many logistic problems in terms of road conditions, availability of suitable vehicle available for inspections, tracing cases and information flow. A consolidated collaboration between local health Authorities is fundamental for sharing resources and skills.
Au: the sentences above have been inserted at line 198.
I hope that responses have been adequate and have met your requests.
Kind Regards
Fabrizia Guidi

Reviewer 2 Report
The authors describe characterization of a staphylococcal food poisoning outbreak during post-earthquake reconstruction. Identification of S. aureus and detection of staphylococcal enterotoxins as the food poisoning causative reagent are presented.
Comments:
Line 39-43
Several immunological assays for newly identified SE detection have already been reported.
Even in SEs excluding SEA-SEI, some toxins are identified as being involved in staphylococcal food poisoning outbreaks.
Line 78-79
I could not understand the denominator of the percentage of symptoms. Does the method have a description of the calculation method?
Line 91
Vertical lines are unnecessary. Horizontal line is missing in the right end column. Decimal separators are not unified. There are several cells that seem to be missing%.
Line 284
Why did you adopt the multiplex PCR that can detect only 11 SE genes despite the presence of more SE gene detectable multiplex PCR?
Author Response
Dear Reviewer,
Authors would like to thank you for all useful comments and constructive suggestions, which help to improve the quality of this manuscript.
I have listed below, point-by-point, all changes and additions responding to your comments. The revisions on the manuscript have been clearly highlighted using different colors for each Reviewer and in particular red for Reviewers 1, blue for Reviewer 2 and green for Reviewer 3.
As requested by the Editor, the Key Contribution has been included in the revised version.
Point 1: Line 39-43. Several immunological assays for newly identified SE detection have already been reported.
Even in SEs excluding SEA-SEI, some toxins are identified as being involved in staphylococcal food poisoning outbreaks.
Response 1: Authors agree with the observation of the Reviewer 2. Several immunological assays has been described for detection of newly identified SEs. However, these published detection assays dedicated to some new SEs were developed in the frame of research projects and, in general, were tested on culture (eg BHI). In addition, at our knowledge, these methods are not dedicated to food matrices and not available in the commerce. One exception, Hait et al 2018 in collaboration with Biomerieu (Hait JM, Nguyen AT, Tallent SM. Analysis of the VIDAS® Staph Enterotoxin III (SET3) for Detection of Staphylococcal Enterotoxins G, H, and I in Foods. J AOAC Int. 2018 Sep 1;101(5):1482-1489. doi: 10.5740/jaoacint.17-0501. Epub 2018 Apr 20) published work on new detection kit for toxins SEG SEH and SEI, we are involved in this work as French NRL but this kit will not be available in the commerce this year (2019). Finally, in our role of official laboratory, NRL and EURL, methods must be validated before their implementation in our activities. Thus, The European screening method (ESM) of European Reference Laboratory for CPS is the only one cited as a reference method in the criterion 1.21 for SEs by EC Regulation 2073/2005 modified (Annex I, Chapter 1). This method is used for own checks and official controls in the EU Member States.
However, as discussed at EURL/NRLs workshops and as reported by EFSA, it is necessary to develop official detection tools targeting SEG, SEH, SEI. EURL for CPS work on an ELISA-based technique allowing the quantification of SEG, SEH and SEI, that could be transferred to NRLs. This work was presented in FoodMicro congress in Berlin 2018 (see poster in attached).
The Authors acting as Competent Authorities must comply with the current legislation and their laboratories only adopt the official method until the EURL provides new assays.
Point 2: Line 78-79. I could not understand the denominator of the percentage of symptoms. Does the method have a description of the calculation method?
Au: Agreed and done.
Response 2: Authors agree with the reviewer 2, it would be better to specify the denominator of the percentage that corresponds to the number of affected workers (cases). The information has been inserted at line 78 : “Among the affected workers (n=26)…”
Point 3: Line 91. Vertical lines are unnecessary. Horizontal line is missing in the right end column. Decimal separators are not unified. There are several cells that seem to be missing%.
Au: Agreed and done.
Response 3: Vertical lines have been removed from Table 1. Horizontal line has been modified. Decimal separators have been unified adopting for all numbers the decimal point. As for the percentages, the numbers without % indicate the Relative Risk that is (the ratio between the incidence of cases in exposed group and incidence of cases in unexposed group) a dimensionless number.
Point 4: Line 284. Why did you adopt the multiplex PCR that can detect only 11 SE genes despite the presence of more SE gene detectable multiplex PCR?
Response 4: At the time of the outbreak, it was known that SEG, SEH and SEI, SEM, SEN, SEO, SER, SES, and SET have shown to be emetic (Argudin et al. /Toxins 2010, 2, 1751-1773; doi: 10.3390/toxins2071751; D.L. Hu, A. Nakane / European Journal of Pharmacology 722 (2014) 95–107; S. Johler et al./ Toxins 2015, 7(3), 997-1004; doi:10.3390/toxins7030997). The multiplex PCR panel provided by the EURL for CPS in addition to targeting the classical SEs’s genes, also detects seg, seh, sei, sep sej and ser because the corresponding SEs are the main for which an emetic activity has been demonstrated. Also, EURL for CPS is developing Multiplex PCR dedicated to SEQ, SEK and SEU. Tests performed on 62 strains issued from SFPO showed the presence of these genes. This work is at the validation step and will be published later. Another work (PhD) on PCR dedicated to genes SEM, SEN and SEO is in progress and showed the presence of these genes in more than 50 strains incriminated in several SFPO.
By other hand, EURL for CPS implement WGS method. WGS methods allow access to a large amount of information on the sequenced strain. Oral presentation and one poster were presented at the IAFP symposium (April 2018, Stockholm) on
(i) Oral: the Comparison of typing reference methods and Whole Genome Sequencing (WGS) analyses for the characterization of S. Aureus strains isolated from food outbreaks. This presentation showed the capability of the new tool to detect 23 CPS genes after application on 143 genomes corresponding to SFPO in Europe from 2005 to 2017 and some reference strains.
(ii) Poster Genetic diversity of staphylococcal strains isolated from food and enterotoxin coding genes (see poster in attached). This work showed the development of Genomic research allows to detect all SE in 143 strains
This WGS will be published by the beginning of 2019. By the way, Authors have included the S. aureus of this poisoning in a research project at the beginning managed by the Italian National Reference Laboratory for CPS and consisting of Whole Genome Sequencing of the strains isolated from different outbreaks. The complete genome sequence will provide the full toxigenic profile of the strains.
I hope that responses have been adequate and have met your requests.
Kind Regards
Fabrizia Guidi

Reviewer 3 Report
This article present an investigation of foodborne outbreak in central Italy, that involved 26 workers employed in the post-earthquake reconstruction. This retrospective cohort study identified a pasta salad as the most likely source of food poisoning. Microbiological and molecular typing analysis identified Staphylococcus aureus and associated toxin as the most likely cause of the outbreak.
Please see below my comments and suggestions to improve the manuscript:
Line 124: Given the decrease in price of whole genome sequencing of bacterial genome, the author should consider to assess the S. aureus strain by whole genome sequencing. This will provide the full toxigenic profile of the strains together with highly accurate typing information on the strain suspected to cause the food poisoning outbreak. These data would be valuable for similar future investigations. Assessing the sequence type (MLST) profile of the S. aureus isolates would also be valuable.
Line 127: Could the authors present the PFGE profile of the isolates and PCR results of sea, sed, sej and ser genes. These results should be presented to support the conclusion that contamination had the same clonal origin (Line 190).
Line 206: Could the authors provide references of such successful investigation.
Author Response
Dear Reviewer,
Authors would like to thank you for all useful comments and constructive suggestions, which help to improve the quality of this manuscript.
I have listed below, point-by-point, all changes and additions responding to your comments. The revisions on the manuscript have been clearly highlighted using different colors for each Reviewer and in particular red for Reviewers 1, blue for Reviewer 2 and green for Reviewer 3.
As requested by the Editor, the Key Contribution has been included in the revised version.
Point 1: Line 124: Given the decrease in price of whole genome sequencing of bacterial genome, the author should consider to assess the S. aureus strain by whole genome sequencing. This will provide the full toxigenic profile of the strains together with highly accurate typing information on the strain suspected to cause the food poisoning outbreak. These data would be valuable for similar future investigations. Assessing the sequence type (MLST) profile of the S. aureus isolates would also be valuable.
Response 1: Authors totally agree with the Reviewer 3 on the importance of a further typing study on the strains. EURL for CPS implemented WGS method. WGS methods allow access to a large amount of information on the sequenced strain. Oral presentation and one poster were presented at the IAFP symposium (April 2018, Stockholm) on
(i) Oral: the Comparison of typing reference methods and Whole Genome Sequencing (WGS) analyses for the characterization of S. Aureus strains isolated from food outbreaks. This presentation showed the capability of the new tool to detect 23 CPS genes after application on 143 genomes corresponding to SFPO in Europe from 2005 to 2017 and some reference strains.
(ii) Poster Genetic diversity of staphylococcal strains isolated from food and enterotoxin coding genes. This work showed the development of Genomic research allows to detect all SE in 143 strains
This WGS will be published by the beginning of 2019. By the way Authors have included the S. aureus of this poisoning in a research project at the beginning managed by the Italian National Reference Laboratory for CPS which will consist of Whole Genome Sequencing of the strains isolated from different outbreaks.
Point 2: Line 127: Could the authors present the PFGE profile of the isolates and PCR results of sea, sed, sej and ser genes. These results should be presented to support the conclusion that contamination had the same clonal origin (Line 190).
Au: Agreed and done.
Response 2: The Image of the Agarose gel electrophoresis patterns showing the amplification of sea, sed, sej and ser genes (Fig.1) has been inserted at line 127. The UPGMA dendrogram (Fig. 2) illustrating similarity of PFGE profiles has been inserted at line 129
Point 3: Line 206: Could the authors provide references of such successful investigation.
Au: Agreed and done.
Response 3: Line 206: if what the Reviewer 3 requires are some examples of successful investigation with the identification of the involved food, the Authors inserted two references in this regard.
I hope that responses have been adequate and have met your requests.
Kind Regards
Fabrizia Guidi

Round 2
Reviewer 1 Report
I thank the authors for responding to each of my previous questions and comments well. I look forward to a report in the future on the whole genome sequencing of the causative MSSA strain.
I have just 2 comments on the revised manuscript:
1. The manuscript is readable, but it will require substantial language editing by a native speaker. The awkward English language in the manuscript does not interfere with the ability of a reader to understand the intended meaning of the authors, but it is distracting.
2. Line 52. "FPO" should presumably be "FBO". I apologize if this was in the previous manuscript and I did not see it.
Thank you for the opportunity to review this revised manuscript.
Author Response
Response to Reviewer 1 Comments
Point 1: The manuscript is readable, but it will require substantial language editing by a native speaker. The awkward English language in the manuscript does not interfere with the ability of a reader to understand the intended meaning of the authors, but it is distracting.
Response 1: Authors would to thank the Reviewer for this suggestion. The manuscript has been sent to the MPDI Editing Service for a language editing.
Point 2: Line 52. "FPO" should presumably be "FBO". I apologize if this was in the previous manuscript and I did not see it.
Response 2: The abbreviation FPO stands for Food Poisoning outbreak. It is explained at line 41.

Reviewer 2 Report
The manuscript has been revised well. I think this manuscript will be acceptable after a correction has been done.
Line 95. Table 1
Authors should be write a horizontal line under the column of "FOODS, case, Total AR, ...Lower, and Upper"
Author Response
Point 1: Line 95. Table 1
Authors should be write a horizontal line under the column of "FOODS, case, Total AR, ...Lower, and Upper"
Response 1: Agreed and done
As suggested by Reviewer1, the manuscript has been sent to the MPDI Editing Service for a language editing.

Reviewer 3 Report
Dear authors,
Thanks for having incorporated some of the previous suggestions to the manuscript.
Please see below my point by point comments to your response:
Point 1:
If I understand correctly, the authors would like to keep the whole genome sequence of the S. aureus isolates responsible for this food poisoning outbreak for a publication that will be submitted early next year.
I still think that the WGS should be incorporated to the present publication to provide an accurate typing of the isolates together with the full repertoire of toxins. This will allow to precisely track potential new food outbreaks caused by this clone in the future.
Point 2:
The addition of PFGE and multiplex PCR for the detection support the conclusion about the clonal origin of the strains causing the intoxication. Nevertheless, figure 1 could be better annotated to facilitate interpretation.
Best regards,
Author Response
Response to Reviewer 3 Comments
Point 1: If I understand correctly, the authors would like to keep the whole genome sequence of the S. aureus isolates responsible for this food poisoning outbreak for a publication that will be submitted early next year.
I still think that the WGS should be incorporated to the present publication to provide an accurate typing of the isolates together with the full repertoire of toxins. This will allow to precisely track potential new food outbreaks caused by this clone in the future.
Response 1: Dear Reviewer,
Authors agree that the WGS represent an unavoidable tool allowing to precisely track potential new food outbreaks caused by this clone in the future. In this way, authors from both EURL for CPS and Italian reference network are working on the implementation of the tool and to study more than 143 strains (and 97additional strains in progress) issued from Staphylococcal Food Poisoning Outbreaks (SFPO) in Europe. Strains issued from this study will be included in our future publication, but the results are not available in the moment. All our publications concerning SFPO over the period 2005 to 2017 will be referred in our future publication.
Point 2: The addition of PFGE and multiplex PCR for the detection support the conclusion about the clonal origin of the strains causing the intoxication. Nevertheless, figure 1 could be better annotated to facilitate interpretation.
Response 2: Authors tried to make the Figure 1 and its caption more easily interpretable.
Figure 1. Multiplex PCR for SEs genes: results.
1a. Detection of sea, seb, sec, see, and ser genes: lane 1: negative control; lanes 2–9: samples; lanes 10, 11, 12, and 13: positive controls for sec (451 bp), see (213 bp), sea (102 bp), seb (164 bp), sed (278 bp), and ser (123 bp).
1b. Detection of seg, seh, sei, sej, and sep genes: lane 1: negative control; lanes 2–10: samples; lane 11: positive control for seg (198 bp), seh (173 bp), and sei (328 bp); lane 12: positive control for seg, seh, sej (131 bp), and sep (396 bp). M, 100-bp size Marker.
As suggested by Reviewer1, the manuscript has been sent to the MPDI Editing Service for a language editing.
